# Effects of Self-Lubricant Coating and Motion on Reduction of Friction and Wear of Mild Steel and Data Analysis from Machine Learning Approach

**DOI:** 10.3390/ma14195732

**Published:** 2021-09-30

**Authors:** Nayem Hossain, Mohammad Asaduzzaman Chowdhury, Abdullah Al Masum, Md. Sakibul Islam, Mohammad Shahin, Osama M. Irfan, Faramarz Djavanroodi

**Affiliations:** 1Department of Mechanical Engineering, International University of Business Agriculture and Technology, Dhaka 1707, Bangladesh; 2Department of Mechanical Engineering, Dhaka University of Engineering and Technology, Dhaka 1707, Bangladesh; asadzmn2014@yahoo.com (M.A.C.); 09masumduetme@gmail.com (A.A.M.); shahin.duet37@gmail.com (M.S.); 3Department of Computer Science and Engineering, International University of Business Agriculture and Technology, Dhaka 1707, Bangladesh; sakibul.islam@iubat.edu; 4Department of Mechanical Engineering, College of Engineering, Qasim University, Buraydah 52571, Saudi Arabia; o.ahmed@qu.edu.sa; 5Mechanical Engineering Department, Prince Mohammad Bin Fahd University, Al Khobar 31952, Saudi Arabia

**Keywords:** tribology, mild steel, coating, reciprocating motion, circular motion, morphology

## Abstract

The applications of coated mild steels are gaining significant attention in versatile industrial areas because of their better mechanical properties, anticorrosive behavior, and reproducibility. The life period of this steel reduces significantly under relative motion in the presence of friction, which is associated with the loss of billion-dollar every year in industry. Productivity is hampered, and economic growth is declined. Several pieces of research have been conducted throughout the industries to seeking the processes of frictional reduction. This study is attributed to the tribological behavior of electroplated mild steel under various operating parameters. The efficiency of commercial lubricant and self-lubrication characteristics of coated layer plays a significant role in the reduction of friction. The reciprocating and simultaneous motion in relation to pin as well as disc are considered during experimentation. The lubricating effects in conjunction with motions are responsible for compensating the friction and wear at the desired level. During frictional tests, the sliding velocity and loads are changed differently. The changes in roughness after frictional tests are observed. The coated and rubbing surfaces are characterized using SEM (Scanning Electron Microscopy) analysis. The coating characteristics are analyzed by EDS (Energy Disperse Spectroscopy), FTIR (Fourier-transform Infrared Spectroscopy), and XRD (X-ray diffraction analysis) methods. The lubrication, reciprocating motion, and low velocity result in low friction and wear. The larger the imposed loads, the smaller the frictional force, and the larger the wear rate. The machine learning (ML) concept is incorporated in this study to identify the patterns of datasets spontaneously and generate a prediction model for forecasting the data, which are out of the experimental range. It can be desired that the outcomes of this research will contribute to the improvement in versatile engineering fields, such as automotive, robotics, and complex motion-based mechanisms where multidimensional motion cannot be ignored.

## 1. Introduction

Mild steel is a low percentage of carbon steel alloy [1]. Other elements, such as manganese and silicon, are also present in it [2]. As the carbon percentage increases, this material becomes harder and stronger but becomes less ductile [3]. However, this material has several advantageous properties, including low cost and desirable mechanical properties, and that is why this material is widely used [4]. As mild steels are being continuously used in gears, valves, turbine blades, turbine shafts, and cams, it has become necessary to invent harder and wear-resistive steel components to be utilized in those applications for better service life [5,6,7].

When mild steel is coated by distinct thin film layers, then mechanical and frictional behaviors are enhanced [8,9]. Considering previous studies, it can be said that chromium is coated on different materials to give superior properties to the material [10,11,12]. It exhibits superior properties under the linear motion as well as dry surface finishing process [13]. Nickel is used as a thin film coated layer on different components in the aircraft industry, automobile engineering, and machine and their accessories for better friction and wear properties, anti-corrosive behavior, high hardness, and suitable lubricating characteristics [14,15]. To ensure reasonable tribological properties, copper can be considered a suitable coating material in the industry [16].

Frictional behaviors of different materials are enhanced by combining the knowledge of machine learning (ML) with mechanical engineering. Patterns are automatically detected by the ML methods in datasets, and a model is developed for the prediction of data for further researches [17,18,19].

In practical applications, combined effects of reciprocating and rotating motion are important. Previous studies [20,21] have shown that in the case of rotating motion, friction and wear are seemed to be higher than that of reciprocating motion, and different results are also noted in few cases. However, no clear correlations have been found among combined motion, friction, and wear. In this study, combined motion is inspected to evaluate the trends of tribological behavior. The combined motion is used at the same time in different mechanical systems, robotics components, electronic devices, and similar real-life and industrial applications [22,23,24,25,26,27]. Several pieces of research have been done on friction and wear of different types of materials under different operating and processing conditions with computing approaches [28,29,30,31,32,33,34,35,36,37].

The novelty of this study is the electroplated coating of Cr-Cu-Ni on mild steel, and the coating layer produces self-lubricating characteristics on the surface. The motions in different directions are also incorporated by design in the setup. The variable normal loads, variable sliding velocities, and commercially available lubricant are also applied. Combined effects of all the variables are used differently during experimentation for keeping the frictional force and wear rate as low as possible. The coating layers themselves have advantages for improving mechanical as well as tribological properties in the industry over the materials in which suitable coating layers are absent. The lower friction results in lower consumption of energy. Moreover, the materials losses are reduced in tribological systems, which, in turn, increases the life of the systems and makes them sustainable. In addition, the impacts of these improvements on tribological issues are confirmed sustainable productivity and economic growth. SAE 60 lubricant is chosen because of its better tribological effects, viscosity-temperature relationships, anti-oxide and black sludge formation, and cleaning properties [38]. The types of materials are important parameters that play a significant role in varying the friction differently, even in the same or different conditions. In this study, mild steel under motions, coating, and the lubricating condition is analyzed, whereas, in another study [28], aluminum is considered. Due to the different types of materials, coating characteristics, lubrication process, and affinity between pin and disc, the results are varied significantly. The variations of results can be used as resources for future researchers to design the system differently by using actual and predicted data in the future.

## 2. Methodology

### 2.1. Material Preparation

The disc for the experiment is three-layered electroplated mild steel in which chromium, copper, and nickel are used as a coating material.

The mild steel disc is coated by the Jatrabari metal company, Dhaka. Initially, mild steel sheets are collected and cut into disc shape from a machine shop by CNC lathe, and the disc is deposited by coating chromium-copper-nickel particles (Figure 1). The dimension of the circular sample is Φ50 mm × t = 3 mm.

The electrodeposition process is used to deposit chromium (Cr), copper (Cu), and nickel (Ni) on mild steel surfaces. Several variables, such as density of the current, bath temperature, electrolyte composition, pH of the electrolyte, and related parameters, have some role in coating performance [39,40]. The cross-section of the specimen is used to determine the g thickness of Cr-Cu-Ni coating, which is almost 25.24 μm. SS 304 is considered as a pin sample (6 mm diameter) that has a 48 mm height along with a 7 mm clamping length (3 mm diameter). The projection view pin sample is illustrated in Figure 2. Stainless steel chemical components of silicon, chromium, manganese, and iron in percentage are 0.47 ± 0.14, 13.15 ± 0.49, 8.82 ± 0.49, and 77.57 ± 0.66, respectively.

### 2.2. Experimental Procedure

The schematic diagram of the pin-on-disc tribometer is indicated in Figure 3, which is used to measure frictional properties. The Cr-Cu-Ni chemical components are coated on a mild steel disc sample by the electroplating process. Three holes are produced by drilling operation, and these holes are used to clamp the specimens with the rotating table by metal screws. In the case of a pin, reciprocating and simultaneous motions are considered, while in case the of disc materials, only simultaneous motion is considered during experimentation. The tests are conducted at ambient temperature and air pressure.

All the experiments are conducted in either dry or lubricated conditions. The total number of experiments is forty-eight, of which twenty-four experiments are done for reciprocating pin motion, and the other twenty-four experiments are done for pin and disc simultaneous motion. Among them, twelve tests are conducted under lubricating conditions. The duration of each test is fifteen minutes, and data are collected continuously by a digital indicator, which is interfaced with a computer. The sliding velocity for reciprocating motion is 0.15 m/s to 0.25 m/s, while for simultaneous motion, it is 0.35 m/s to 0.45 m/s. Four experiments are performed at the same condition to ensure accuracy. This is due to the differences in experimental results under dry and lubricating conditions, confirming the effectiveness of the lubricant. Before and after each experiment, the weight of the pin and disc is measured with caution by a digital weight meter. The weight for both pin and disc is measured in grams, and 3 digits are considered after the point. From the differences in the experimental data (before and after), the wear loss is found in both disc and pin. The surfaces after the tribological test are shown in Figure 4.

### 2.3. Characterization Approaches

ML method is applied to analyze the differences of the COF under different conditions for forecasting the data for further investigation, and a regression model is developed. The quadratic polynomial model has shown better agreement with the obtained experimental results. A polynomial second-degree quadratic equation is as follows:(1)$$ax^{2}+bx+c=0$$

Moreover, the equation is extended up to the *n*th value using the following formula
(2)$$y=b_{0}+b_{1}x_{1\;}+b_{2}x_{1}^{2}+b_{2}x_{1}^{3}+\cdots b^{n}x_{1}^{n}$$

Figure 5 shows the step-by-step machine learning approach for predicting the future data, which are out of the experimental range.

The rubbing surfaces of the disc after tribology tests are characterized by a scanning electron microscopy test. The different elemental presence in the disc material is analyzed with the help of energy-dispersive X-ray analysis. The crystallographic structure is analyzed by the XRD method. FTIR test is conducted to find the existence of bands and fictional groups in materials investigated.

The diagrams of this research paper are drawn using Origin pro, Python framework, AutoCAD, and Microsoft PowerPoint. These software are user-friendly, fast, and make diagrams clear.

## 3. Results and Discussion

### 3.1. Analysis of the Coating Layer on the Material Investigated

#### 3.1.1. Effects of Coating

Self-lubricating coatings are one of the strategies that help to lower the friction values and wear of a surface (Figure 6). Low shear strength is exhibited by the interfacial surfaces as self-lubricating coatings are deposited on the substrate, causing interlayer sliding. Low adhesion results in low shear strength between pin material and coated surface, reducing friction and wear. Some influential parameters, such as relative hardness coatings, the coating thickness, substrate and lubricity, and surface roughness, influence the variation of tribological properties [41,42,43].

#### 3.1.2. Roughness Variation of Coating Surface

Roughness variation of electroplated coating samples before the frictional investigation can be seen in Figure 7. The range along the x-axis is 7.5 mm. The range of the roughness test of the sample is based on the surface profile. Ra value of the coating surface is seen as only 0.40 µm when the friction test is not done.

#### 3.1.3. Scanning Electron Microscopy Analysis of the Coated Sample

Figure 8 shows clean and dirt-free pictures from the samples after the scanning electron microscopy test shows clean and dirt-free pictures (Figure 8). Crack and pore in micro-level and other defects are not visible from any of the pictures.

#### 3.1.4. EDX of Coated Sample

Figure 9 indicates the full area EDX micrographs of the mild steel sample, which is deposited by Cr-Cu-Ni. Ni is the maximum composition contained by the disc. The presence of other elements is also seen in the EDX spectra, such as Cr and Cu. Cr and Cu show a comparatively weak peak.

#### 3.1.5. X-ray Diffraction Analysis of the Coated Sample

X-ray diffraction nature of coating-adhered mild steel is illustrated in Figure 10. Phases show consistency in relation to the image of the scanning electron microscopy and energy-dispersive X-ray findings. The crystal structure of the sample is confirmed by the presence of the peaks.

#### 3.1.6. Fourier Transform Infrared Ray Analysis of the Coated Sample

Figure 11 does not show any peaks in the FTIR analysis. A homogeneous line of bands C–O, C=C, C=O, C–H, O–H does not show any variation or chemical reaction with respect to the property of the materials.

#### 3.1.7. Adhesiveness of the Coating

The cross-hatch adhesion testing method has been used with the ASTM3359 standard to examine the adhesion performance of the thin film on mild steel. Scratching on the coated surface and peeling operation are done using adhesion tape. The range of the adhesion scale is maintained within 5 B to 1 B, in which 5 B is considered as hundred percent adhesion, while 1 B is considered as low adhesion (above sixty percent deposition on test sample) [44]. The output obtained from the electrodeposition method suggests the coating adhesion quality in the range 5 B (100%) and 4 B (90%).

#### 3.1.8. Modulus of Elasticity and Hardness Analysis

The tensile test is conducted for mild steel and electroplated mild steel. The in-plane tensile results indicate that the modulus of elasticity is 203.40 GPa for mild steel and 212.34 GPa for electroplated mild steel. The data collected from the tensile test have ensured that there are lower coating stiffness effects on the film stiffness, and small numbers of tangible deviations are realized [45,46].

The Brinell hardness number for mild steel is found to be 127. But after coating Cr, Cu, Ni on mild steel, it is 141. This result signifies that electroplated layers enhance the hardness of mild steel, which is noticeably influenced by Cr, Cu, and Ni.

### 3.2. Lubricant Effect

#### 3.2.1. COF Analysis

The properties of the lubricant reduce the COF and wear rate in all the experiments. At 3.5 N normal load, 0.45 m/s disc velocity, 0.15 m/s pin velocity, the effect of the lubricant can be seen in Figure 12 for both simultaneous and reciprocating motion. The findings show that for reciprocating and simultaneous motion, the lubricated condition has minimum friction. At the simultaneous motion of pin and disc and at dry condition, the friction factor observed earlier is 0.189, which increases with time and reaches the peak value of 0.197 after 12 min of friction. The COF then declines after 13 min, finally reaching 0.194 [47]. At the dry and reciprocating motion, the initial COF is found to be 0.179, which reaches the highest value of 0.191 after 11 and 12 min and then declines, reaching 0.187 after 15 min of friction. At the simultaneous motion of pin and disc and lubrication, initial friction is observed to be 0.164, which goes up with duration and reaches 0.172 after 8 and 9 min and declines again after 10 min, reaching 0.166 after 15 min. Similarly, at the reciprocating motion of pin and dry condition, the COF is seen to be 0.157 initially, which goes up with time and touches the peak value of 0.165 after 13 min of rubbing, finally declining to 0.162. As tribolayer is formed on the mild steel surface, comparatively less COF is seen in lubricated conditions [48]. When the lubricant is applied, the thin film of the lubrication between the contacting surfaces separates the pin and disc, causing the reduction in contact between the mating surfaces, resulting in lower tribological parameters, as shown in Figure 13. Furthermore, the lubricant makes the roughing surface wet and smoother, lowering the frictional parameters [49,50]. Variation of COF at distinct operating conditions can be seen in Table 1.

In this research work, the ML method has been addressed for further analysis. ML method is frequently used in the field of computer engineering [51], biomedical engineering [52], petroleum engineering [53]. When regression analysis of the table data is done, it results in three separate regression predictions models—blue, red, and green—for each case, as shown in Figure 14. When the experiment parameter changes from simultaneous motion of pin and disc to reciprocating motion of pin at dry condition, the blue regression line shows the comparison of values. Starting from 0.18, the value increases to 0.19, where the noticeable change is −5% to −3%. When the experiment parameter changes from simultaneous motion of pin and disc to reciprocating motion of pin at lubricated condition, the red regression line shows the comparison of values. Starting from 0.18, the value increases to 0.19, indicating upward and downward volatile values. When the experiment parameter changes from simultaneous motion of pin and disc to reciprocating motion of pin at lubricated condition, the green regression line shows the comparison of values. The values start from 0.18 and gradually increase to 0.19. However, the data generated show a gradual increase and decrease trends.

#### 3.2.2. Wear Rate Analysis

The effect of lubricant can be seen in Figure 15 at 3.5 N applied load, 0.45 m/s disc velocity, and 0.15 m/s pin velocity for both simultaneous motion pin and disc and reciprocating motion. The figure confirms the production of less wear at lubricated conditions. Dry condition and simultaneous motion of pin and disc produce a maximum of 0.45 g wear. Compared to the first one, 62.89% less wear (0.167 g) is produced at reciprocating motion and dry conditions. Compared with the first one, 86.67% and 95.55% less wear is produced at the lubricated condition for the simultaneous motion of pin and disc and reciprocating motion of pin, respectively. The produced wears are only 0.06 g and 0.02 g, respectively. Variation of COF at distinct operating conditions can be seen in Table 2.

#### 3.2.3. SEM Analysis

From Figure 16, it is clearly observed that fewer rubbing effects are seen in the lubricated condition. The rubbing effects can be seen at different magnifications from the scanning electron microscopy analysis. No microcrack and microcavity are observable.

#### 3.2.4. Lubrication Regimes Analysis

The lambda ratio (λ) defines the lubrication regime. Lambda ratio is minimal film thickness (*hmin*) with respect to coated surface roughness (Ra1 and Ra2).
(3)$$\lambda =\cdot \lambda =\frac{h_{m_{i}n}}{\sqrt{R_{a_{1}}^{2}+R_{a2}^{2}}}$$

Stribeck states that COF is proportional to the lubricant viscosity, and contact surface speed difference is inversely proportional to the exerted pressure on the contact area. There are three regimes according to Stribeck, namely boundary l, mixed, and hydrodynamic lubrication [54,55,56,57,58].

### 3.3. Effect of Motion

#### 3.3.1. COF Analysis

The pin and disc velocity influence the COF and wear rate in all the experiments shown in Figure 17. The graph is depicted at 2.5 N applied load and dry condition. At 0.45 m/s disc and 0.25 m/s pin velocity, maximum COF is observed. Starting from 0.253, the COF increases and reaches 0.263 after 9 min of rubbing and then declines. At 0.35 m/s disc velocity and 0.15 m/s pin velocity, the starting COF is 0.238, which increases and reaches 0.247 after 7 min and then declines. At reciprocating 0.15 m/s pin velocity, the observed initial COF 0.154 reaches an apex in between 3 and 9 min. At reciprocating 0.25 m/s pin velocity, 0.168 initial COF reaches the highest value after 9 min and then declines to 0.17 after 15 min of rubbing. The second-lowest friction factor is observed to be at 0.25 m/s pin velocity and reciprocating motion. Here, the friction factor goes up initially and then goes down up to a certain time for all conditions due to the formation of a glazed layer [59,60]. The simultaneous motion of pin and disc makes two types of motion, which, in fact, increases surface area, which is responsible for higher friction. Variation of COF at distinct operating conditions can be seen in Table 3.

When regression analysis of the table data is done, it results in three separate regression predictions models—blue, red, and green—for each case shown in Figure 18. When the experiment parameter changes from simultaneous motion of pin and disc to reciprocating motion of pin at dry condition, the blue regression line shows the comparison of values. Starting from 0.25, the value increases to 0.26, indicating continuous increase and decrease. When the experiment parameter changes from higher velocity to lower velocity of pin and disc at dry condition, the red regression line shows the comparison of values. Starting from 0.25, the value increases to 0.26, indicating upward and downward volatile values. When the experiment parameter changes from higher velocity to lower velocity of pin and disc at dry condition, the green regression line shows the comparison of values. The values increase from 0.25 and reach 0.26. However, these data show the gradual increase and decrease trends.

#### 3.3.2. Wear Rate Analysis

The effect of pin and disc velocity for both simultaneous and reciprocating motion at 2.5 N load, at the dry condition in wear rate, is seen in Figure 19. Here, less wear is produced at less pin velocity and reciprocating motion. The highest 0.47 g wear is produced after 15 min of rubbing at 0.45 m/s disc and 0.25 m/s pin velocity, whereas 0.35 g wear occurs at 0.35 m/s disc and 0.15 m/s pin velocity. At the reciprocating pin velocity of 0.25 m/s, 0.17 g wear occurs, whereas 0.15 g wear happens at 0.15 m/s pin velocity. These values are 63.83% and 68.08% lower in comparison to the first value. As the simultaneous motion of pin and disc makes two types of motion, it reaches the larger area of contact level, resulting in large wear. Variation of COF at distinct operating conditions can be seen in Table 4.

#### 3.3.3. SEM Analysis

From Figure 20, it is clearly seen that there are several effects of rubbing and micro cracks. In Figure 20a,d, rubbing effects are observed, whereas, in Figure 20b,c, both rubbing wear track and cracks in micro-level are observed. The results confirm that dry condition harms the surface more in rubbing.

### 3.4. Effect of Normal Load

#### 3.4.1. COF Analysis

Figure 21 shows the comparison of the effect of load in both simultaneous motion of pin and disc and reciprocating motion of pin at 0.45 m/s disc velocity and 0.2 m/s pin velocity and at no lubricated condition. It is found that less friction is observed when more force is applied and at the reciprocating motion of the pin. At 1.5 N load and simultaneous motion, more COF is observed. Initially, the coefficient of friction is observed to be 0.525, which increases with time and attains the peak value of 0.553 after 8 min of rubbing and then declines and reaches 0.535 after 15 min. When 1.5 N load is applied at reciprocating motion, maximum COF is seen as 0.484 after 10 min, which is 0.473 initially and reaches 0.465 after 15 min. At 4.5 N load and simultaneous motion, COF is observed to be 0.205 initially that fluctuates over the period and remains almost the same. At 4.5 N load and reciprocating motion, initial COF is observed as 0.149 that rises to 0.161 after 11 min. Finally, the value of the COF reaches 0.158 [61]. Variation of COF at distinct operating conditions can be seen in Table 5.

Again, the regression analysis of the table data results in three separate regression predictions models—blue, red, and green—for each case shown in Figure 22. When the experiment parameter changes from simultaneous motion of pin and disc to reciprocating motion of pin at dry condition, the blue regression line shows the comparison of values. Starting from 0.52, the value increases to 0.55, indicating continuous increase and decrease. When the experiment parameter changes from less applied load to more applied load at dry condition, the red regression line shows the comparison of values. Starting from 0.52, the value increases to 0.55, indicating upward and downward volatile values. When the experiment parameter changes from less applied load to more applied load at dry condition, the green regression line shows the comparison of values. The values increase from 0.52 and reach 0.55. However, the values change from −71% to −70%.

#### 3.4.2. Wear Rate

The effect of applied load at both reciprocating motion and simultaneous motion in the dry condition is seen in Figure 23. From the figure, it is clearly seen that less wear occurs at lower load and reciprocating motion. After 15 min of rubbing, 0.36 g wear is produced at simultaneous motion and 1.5 N applied load. Comparatively, 55.55% wear happens compared to the first one, which is 0.16 g at reciprocating motion and 1.5 N applied load. Maximum 0.625 g wear is produced, which is 73.61% more in comparison to the first one at simultaneous motion and 4.5 N load. At reciprocating motion and 4.5 N load, 0.24 g wear occurs, which is 33.33% lower in comparison to the first one. The contact area increases when the applied load is increased, which is associated with the contact area, which increases the friction between the mating surfaces, resulting in the formation of more wear. Thus, the applied load increases shear force and accelerates the wear rate. Variation of COF at distinct operating conditions can be seen in Table 6.

#### 3.4.3. SEM Analysis

Figure 24 shows the severe plowing effect and abrasive wear at different magnifications. The surface of the mild steel sample is severely affected by the pin and disc motion. Both plowing effect and abrasive wear are visible at 500 µm, 300 µm, 200 µm, and 100 µm (Figure 24a–d).

### 3.5. Effects of Surface Roughness

The surface roughness of the rubbing surfaces after frictional experimentation is critically observed. The stylus of the roughness checker is moved up to the 7.5 mm length over the rubbing surfaces for measuring the average surface roughness of the specimens. Figure 25a,b shows the higher surface roughness having a Ra value of 1.50 µm and 0.90 µm, respectively. The roughness of 1.50 µm is obtained under the pin and disc simultaneous motion, while the roughness of 0.90 µm is detected under the pin reciprocating motion. Both of these experiments are carried out under dry surface conditions. When motion and dry surface conditions are maintained, then the surfaces become rougher. Due to the roughening effects, the friction and wear are found to be higher. The simultaneous motion exhibits higher frictional values and wearing-out materials than that of the reciprocating motion. This indicates that the surface roughness under reciprocating motion is lower than that of the simultaneous motion. The interesting observation is that the higher the friction and wear, the higher the surface roughness. In the presence of lubrication as well as reciprocating motion, the lower the friction and wear, the lower the surface roughness. The pattern of numerical roughness values is also reflected in SEM surface morphology. Surface asperities, interlocking, surface deformation is responsible for the variation of friction and wear in a very different way. Among them, the surface irregularities result in higher friction when mating surfaces are in contact [48,62]. Surface irregularities are attributed to three mechanisms [63,64]: (i) The surface becomes rougher because of different modes of metal removal rate, (ii) surface texture in the form of waviness is the generated machine-induced vibration or external motion, and (iii) non-controllable parameters are responsible for abnormality and errors. Apart from this, the more the area of contact, the more the friction factor [38]. Under lubrication and simultaneous and reciprocating motion, the surface roughness is 0.70 µm and 0.50 µm, respectively, which is associated with lower friction and wear. These are presented in Figure 25c,d. In this case, lubrication film separates the pin and disc, and the area of contact between the couple surfaces is reduced. This causes lower friction and wear.

### 3.6. Comparison among Surface Roughness, Friction, and Wear

The comparison among COF, wear rate, and surface roughness can be seen in Figure 26 at different conditions. As the contact area reduces for the mating surfaces, less friction and wear are observed in lubrication. Furthermore, the reciprocating motion of pin contacts with less surface area; it also produces less friction and mass loss in comparison to simultaneous motion.

## 4. Conclusions

From this research, it can be concluded that the significant results are the reduction of COF and wear rate during the lubricating condition, reciprocating motion, less pin and disc motion, and lower applied loads. The observed maximum reduction in the COF and wear rate is 62.5% and 95.5%, respectively. The coating on a mild steel surface shows a great effect in reducing friction as well as the wear rate. A smaller number of outliers are exhibited by the regression model, which is developed from the varying COF. Future investigation can be conducted using a different coating material or changing the percentage of coating material.

## Figures and Tables

**Figure 1 materials-14-05732-f001:**
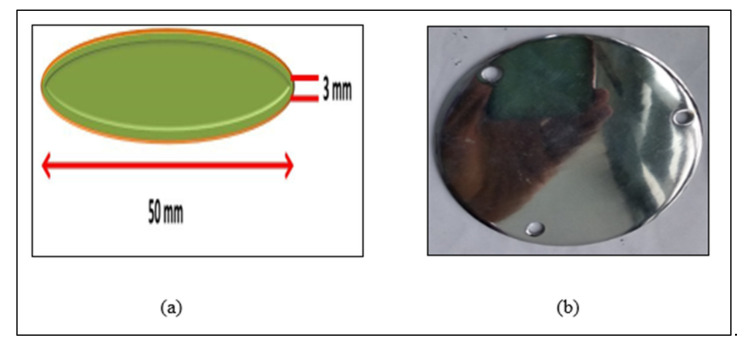
(**a**) Block diagram and (**b**) photograph of disc sample.

**Figure 2 materials-14-05732-f002:**
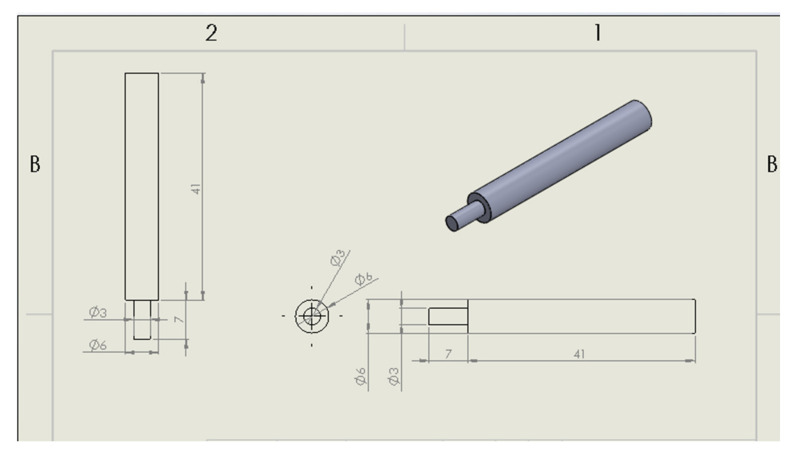
Projection view of pin sample [28].

**Figure 3 materials-14-05732-f003:**
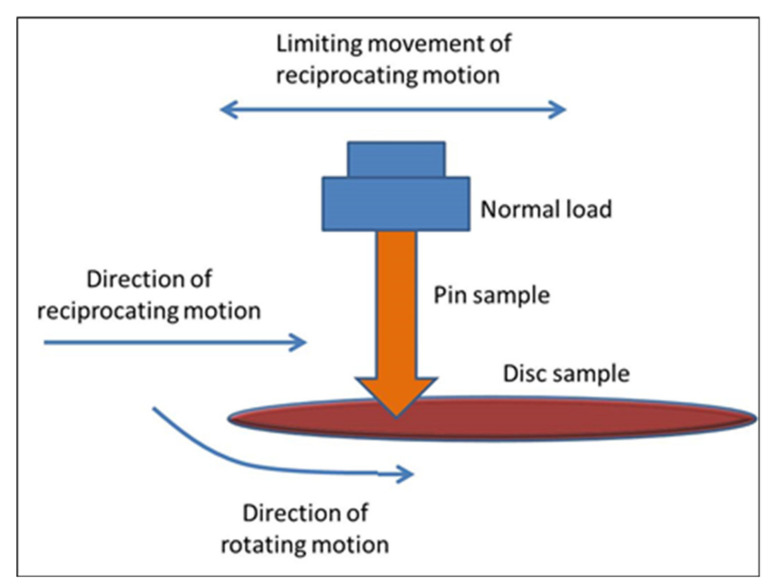
Schematic diagram of the experimental setup.

**Figure 4 materials-14-05732-f004:**
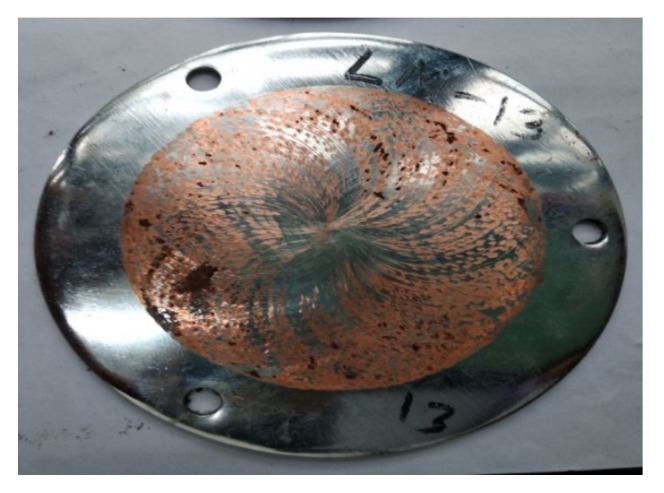
Disc after frictional analysis.

**Figure 5 materials-14-05732-f005:**
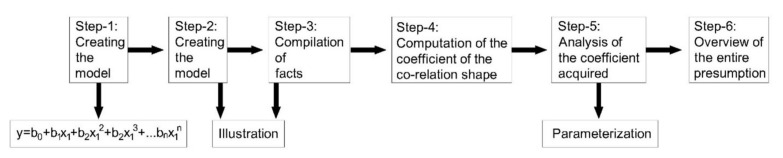
ML approach flow process chart.

**Figure 6 materials-14-05732-f006:**
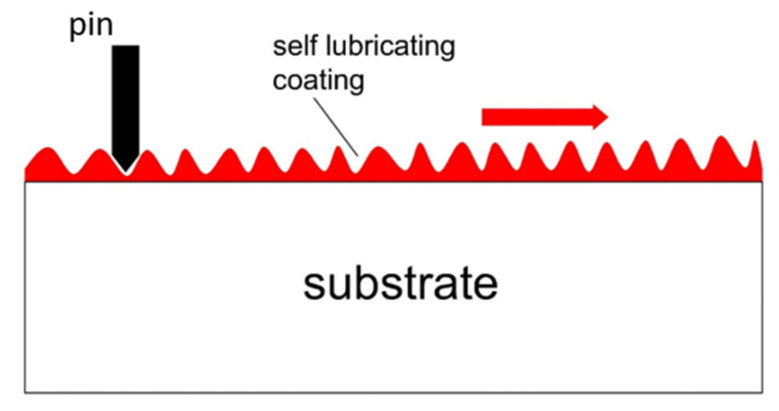
Electroplated coating behavior on the substrate.

**Figure 7 materials-14-05732-f007:**
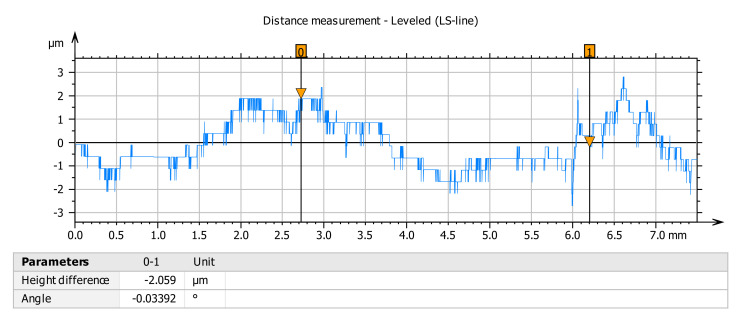
Analysis of the surface roughness of coated mild steel sample before the tribological test.

**Figure 8 materials-14-05732-f008:**
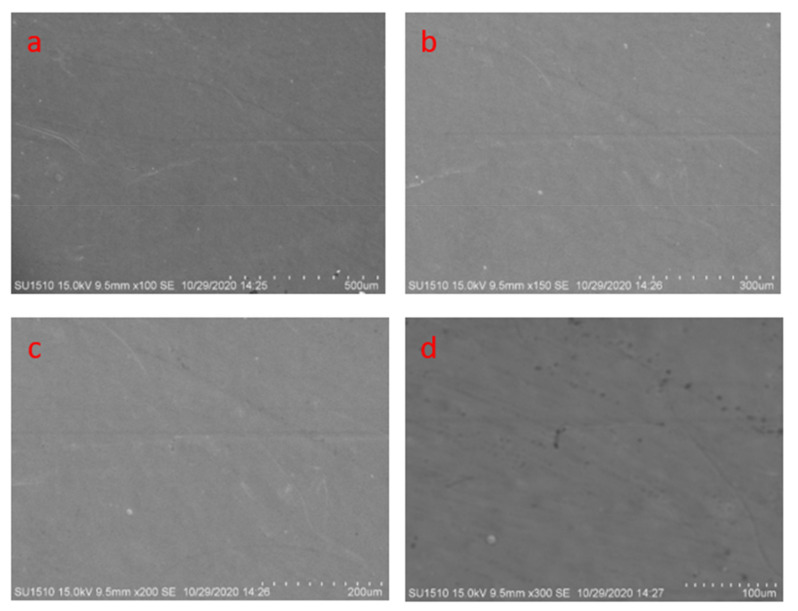
Scanning electron microscopy of coating surfaces at (**a**) 500 µm, (**b**) 300 µm, (**c**) 200 µm, (**d**) 100 µm.

**Figure 9 materials-14-05732-f009:**
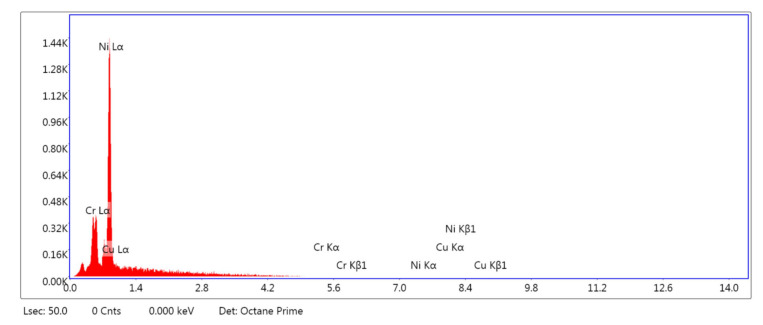
EDX analysis of coated mild steel specimen.

**Figure 10 materials-14-05732-f010:**
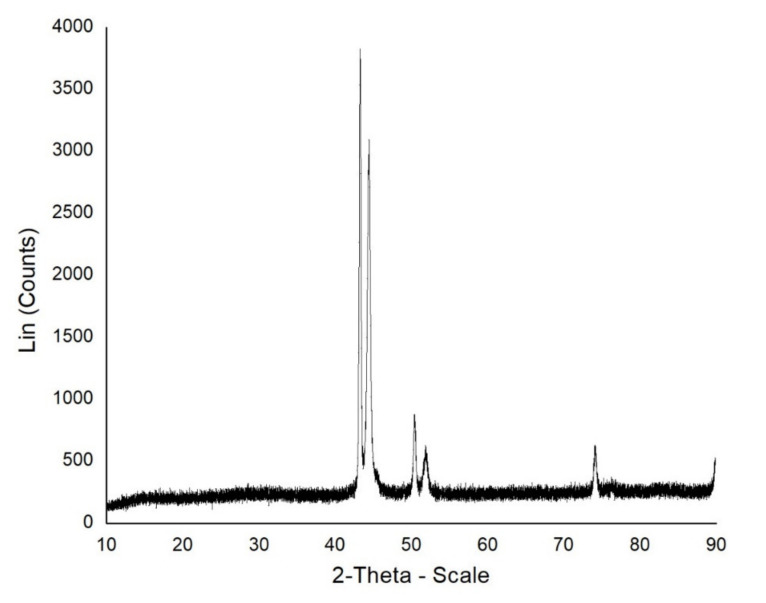
XRD analysis of coated mild steel sample.

**Figure 11 materials-14-05732-f011:**
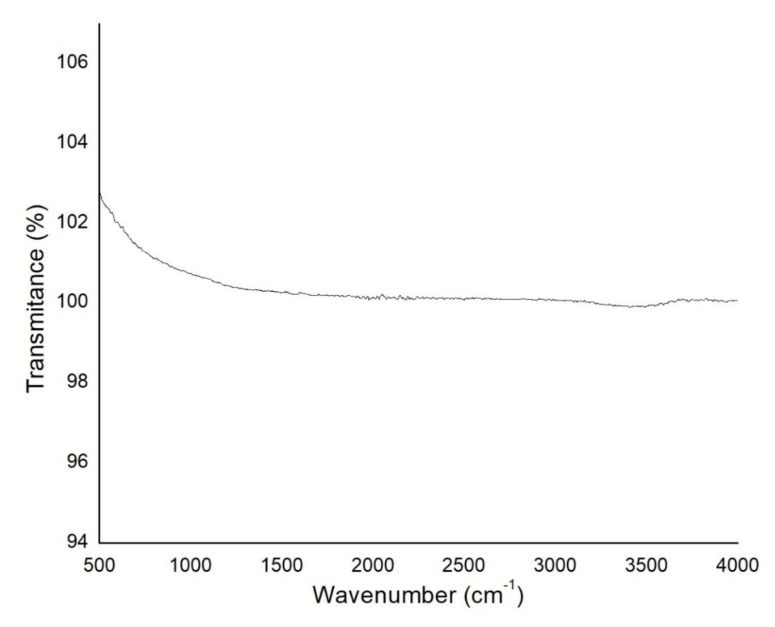
Fourier transform infrared ray analysis of coated mild steel sample.

**Figure 12 materials-14-05732-f012:**
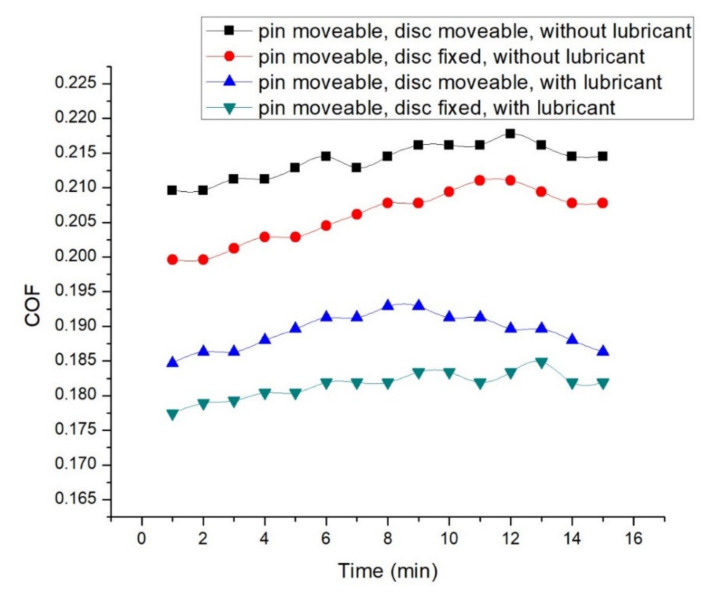
COF at 3.5 N normal load, 0.45 m/s disc, and 0.15 m/s pin velocity.

**Figure 13 materials-14-05732-f013:**
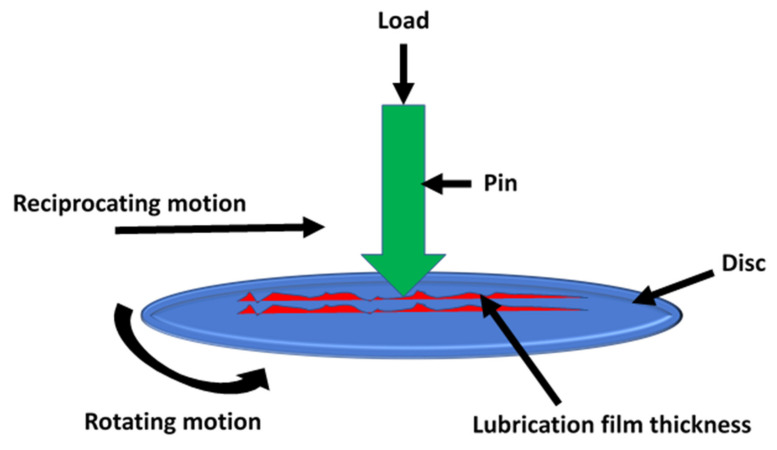
Lubrication mechanism for separating the mating surfaces.

**Figure 14 materials-14-05732-f014:**
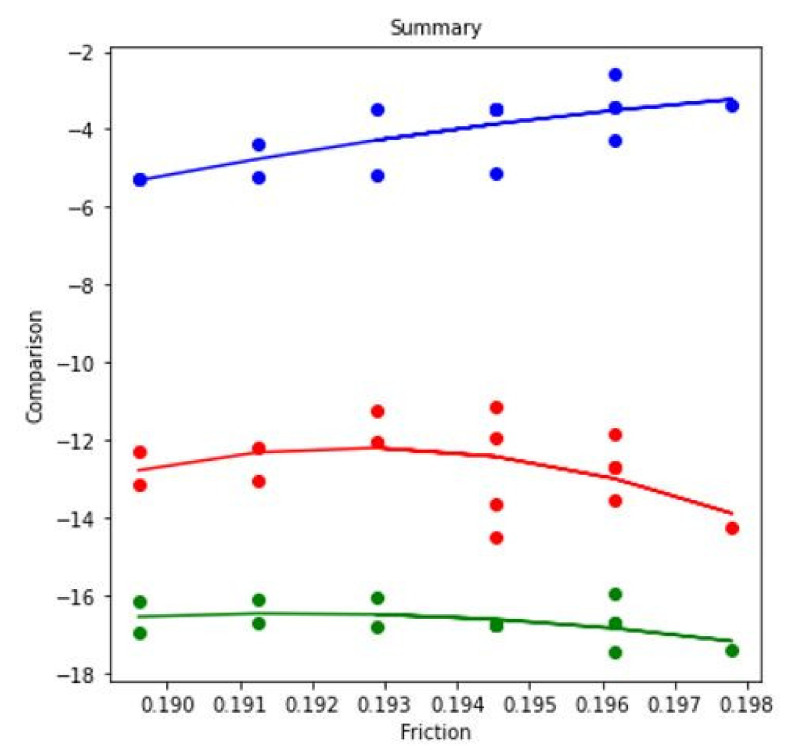
Regression model for COF at 3.5 N applied force, 0.45 m/s disc velocity, and 0.15 m/s pin velocity.

**Figure 15 materials-14-05732-f015:**
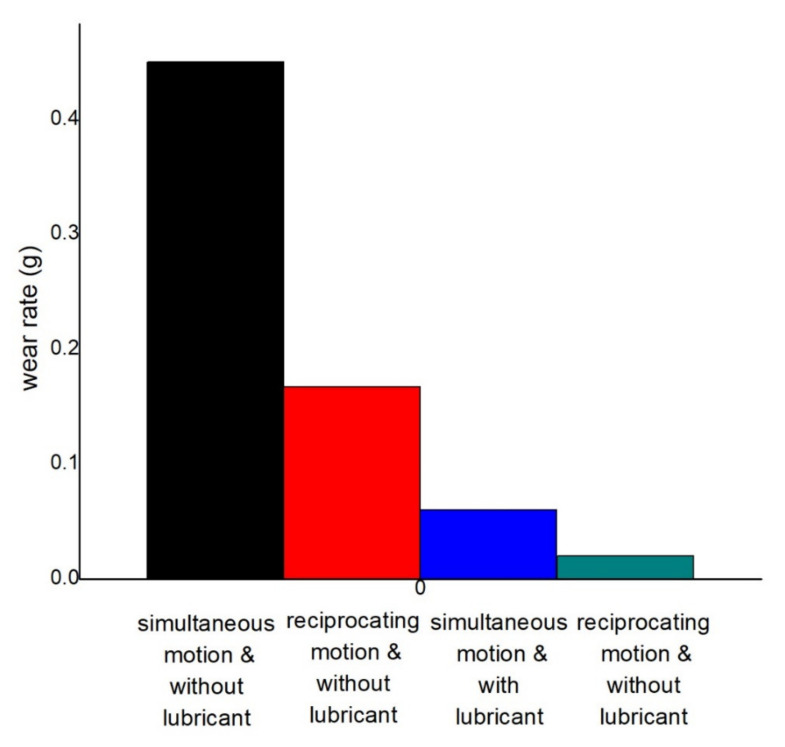
Variation of wear at 3.5 N load, 0.45 m/s disc, and 0.15 m/s pin velocity.

**Figure 16 materials-14-05732-f016:**
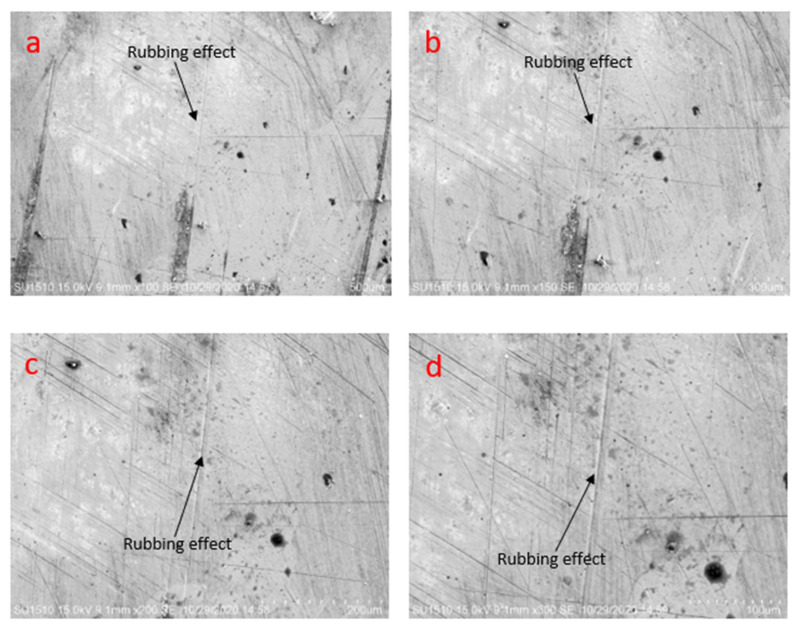
Scanning electron microscopy image of specimens’ later friction test at 3.5 N load, 0.45 m/s disc, 0.15 m/s pin velocity, and lubrication (**a**) at 500 µm, (**b**) at 300 µm, (**c**) at 200 µm, (**d**) at 100 µm.

**Figure 17 materials-14-05732-f017:**
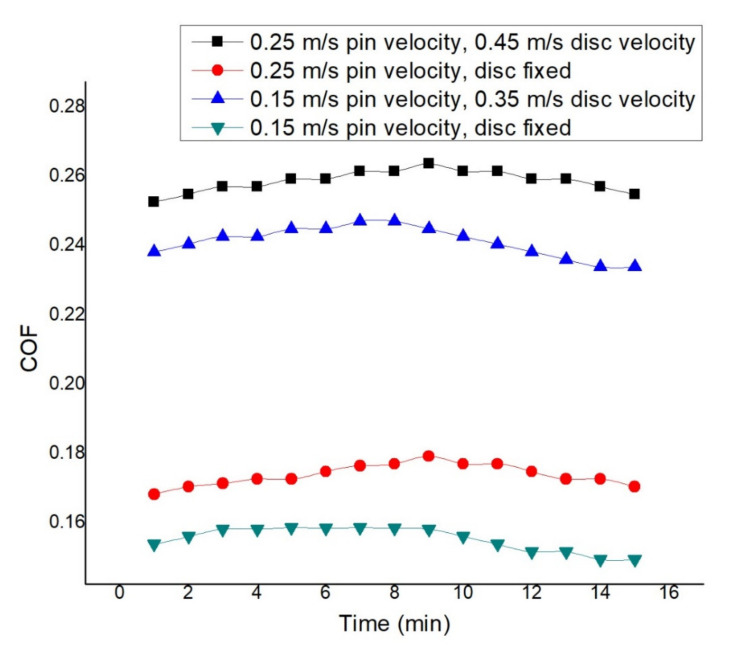
COF at 2.5 N load and dry condition.

**Figure 18 materials-14-05732-f018:**
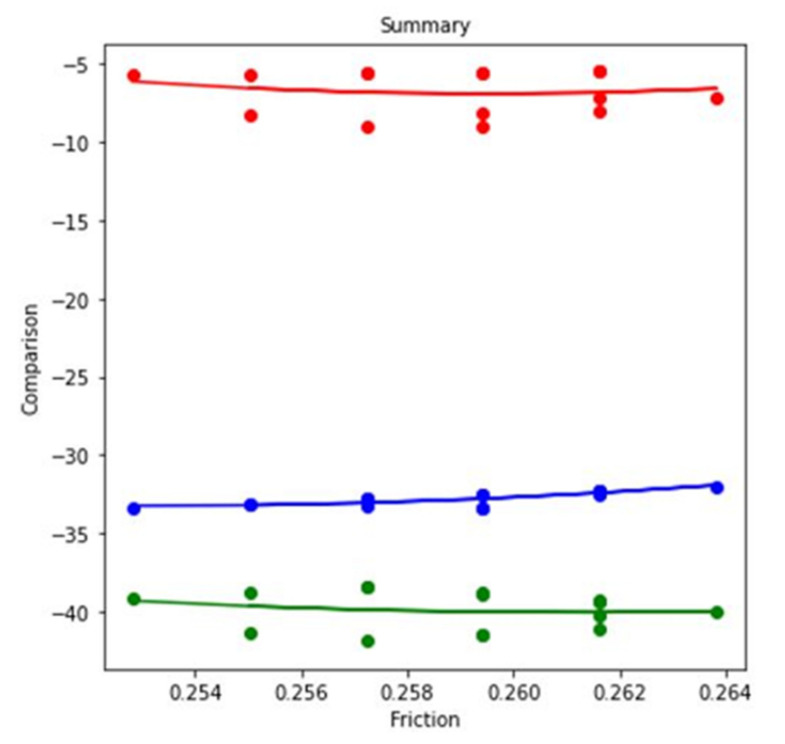
Regression model for COF at 2.5 N load and dry condition.

**Figure 19 materials-14-05732-f019:**
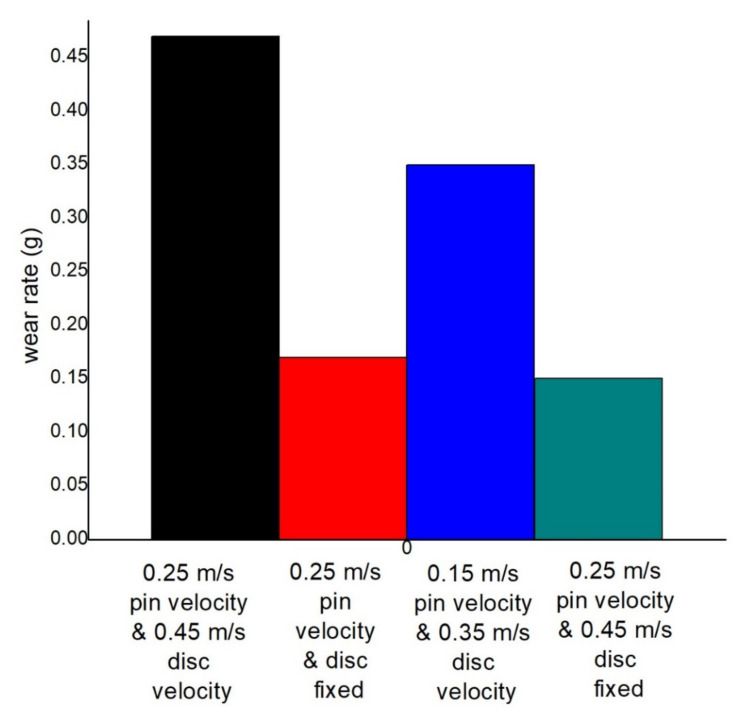
Wear rate comparison at 2.5 N load and dry condition.

**Figure 20 materials-14-05732-f020:**
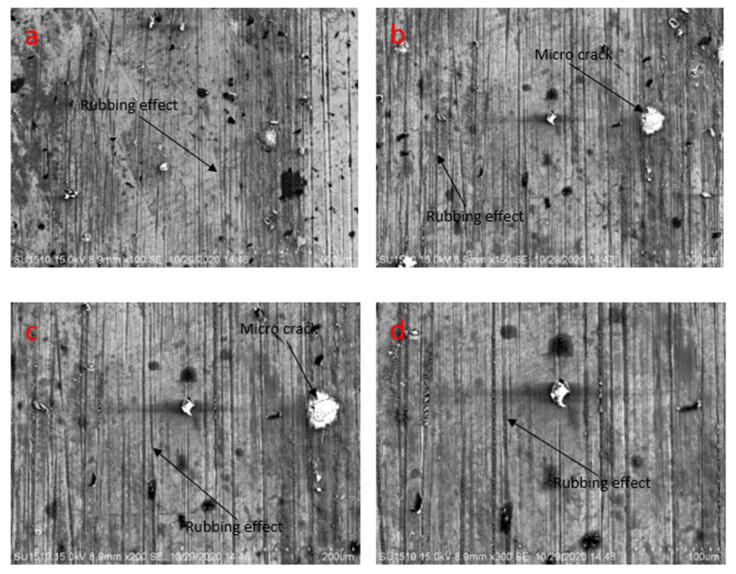
Scanning electron microscopy analysis of the sample after tribological test at 1.5 N applied force, 0.2 m/s pin velocity, and without lubricating condition (**a**) at 500 µm, (**b**) at 300 µm, (**c**) at 200 µm, (**d**) at 100 µm.

**Figure 21 materials-14-05732-f021:**
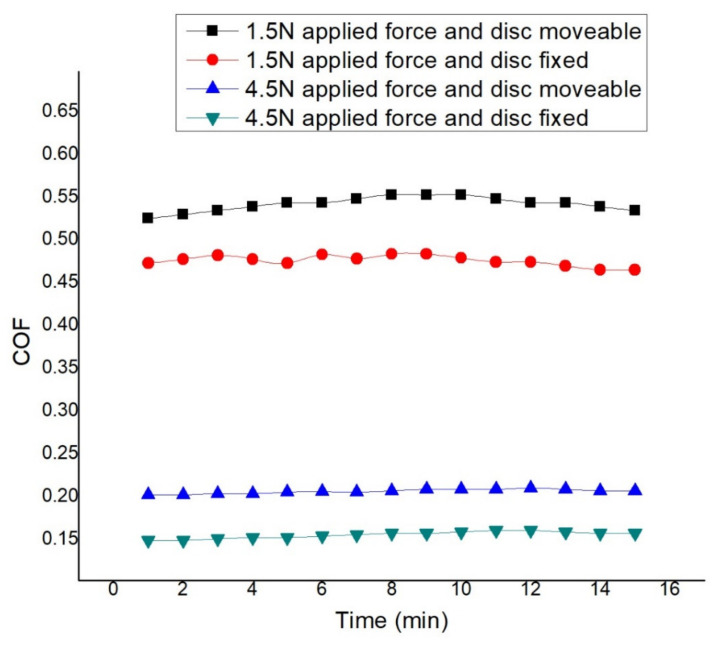
COF at 0.45 m/s disc, 0.2 m/s pin velocity, and dry condition.

**Figure 22 materials-14-05732-f022:**
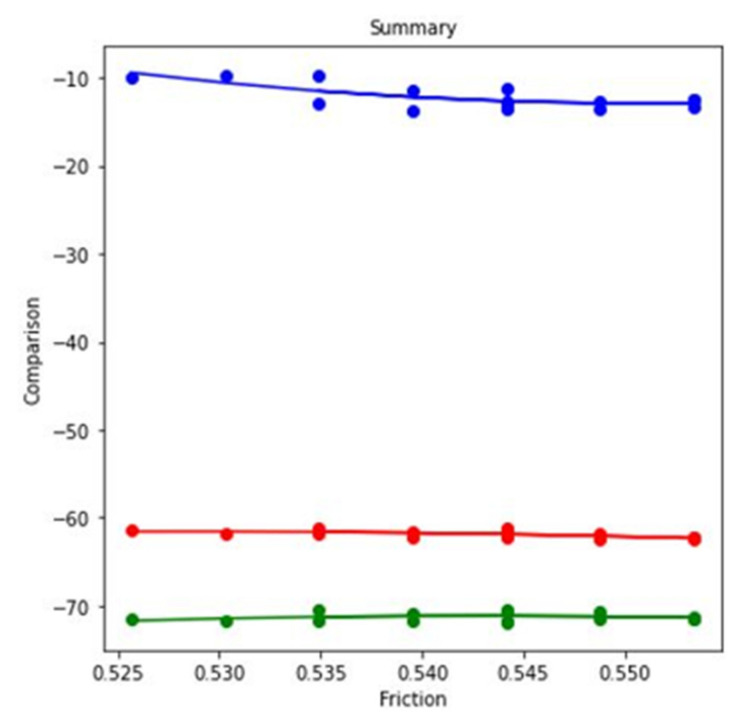
Regression model for COF at 0.45 m/s disc velocity, 0.2 m/s pin velocity, and dry condition.

**Figure 23 materials-14-05732-f023:**
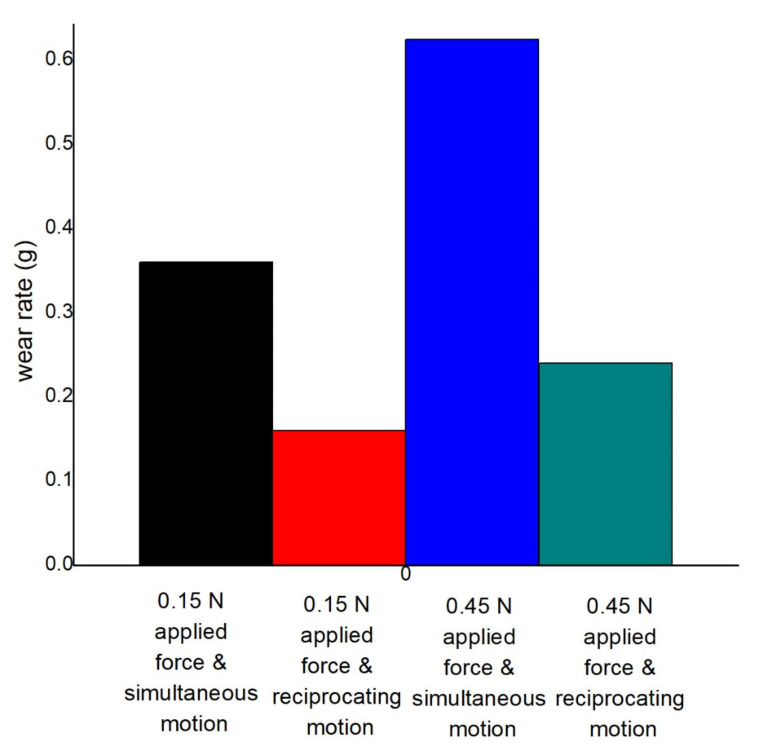
Comparison of wear rate at 0.45 m/s disc, 0.2 m/s pin velocity, and dry condition.

**Figure 24 materials-14-05732-f024:**
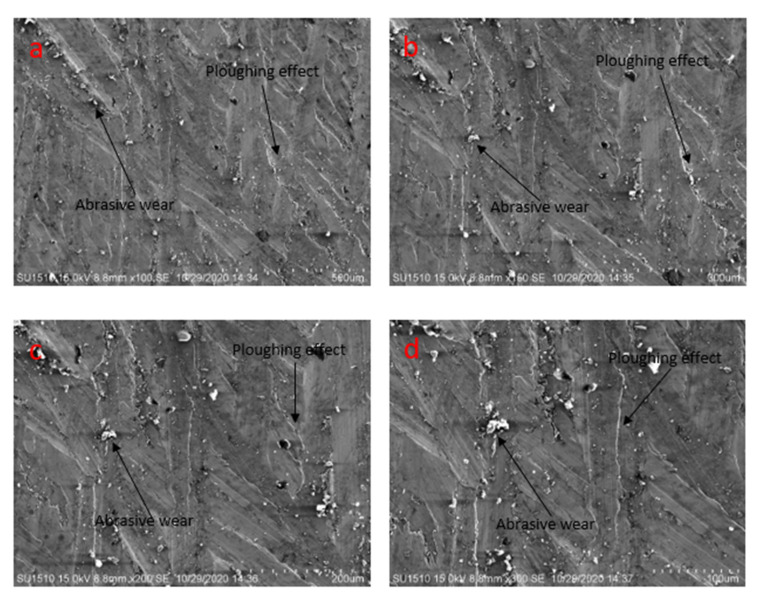
Scanning electron microscopy of the specimens after friction test at 4.5 N load, 0.2 m/s pin, 0.45 m/s disc velocity, and at dry condition (**a**) at 500 µm, (**b**) at 300 µm, (**c**) at 200 µm, (**d**) at 100 µm.

**Figure 25 materials-14-05732-f025:**
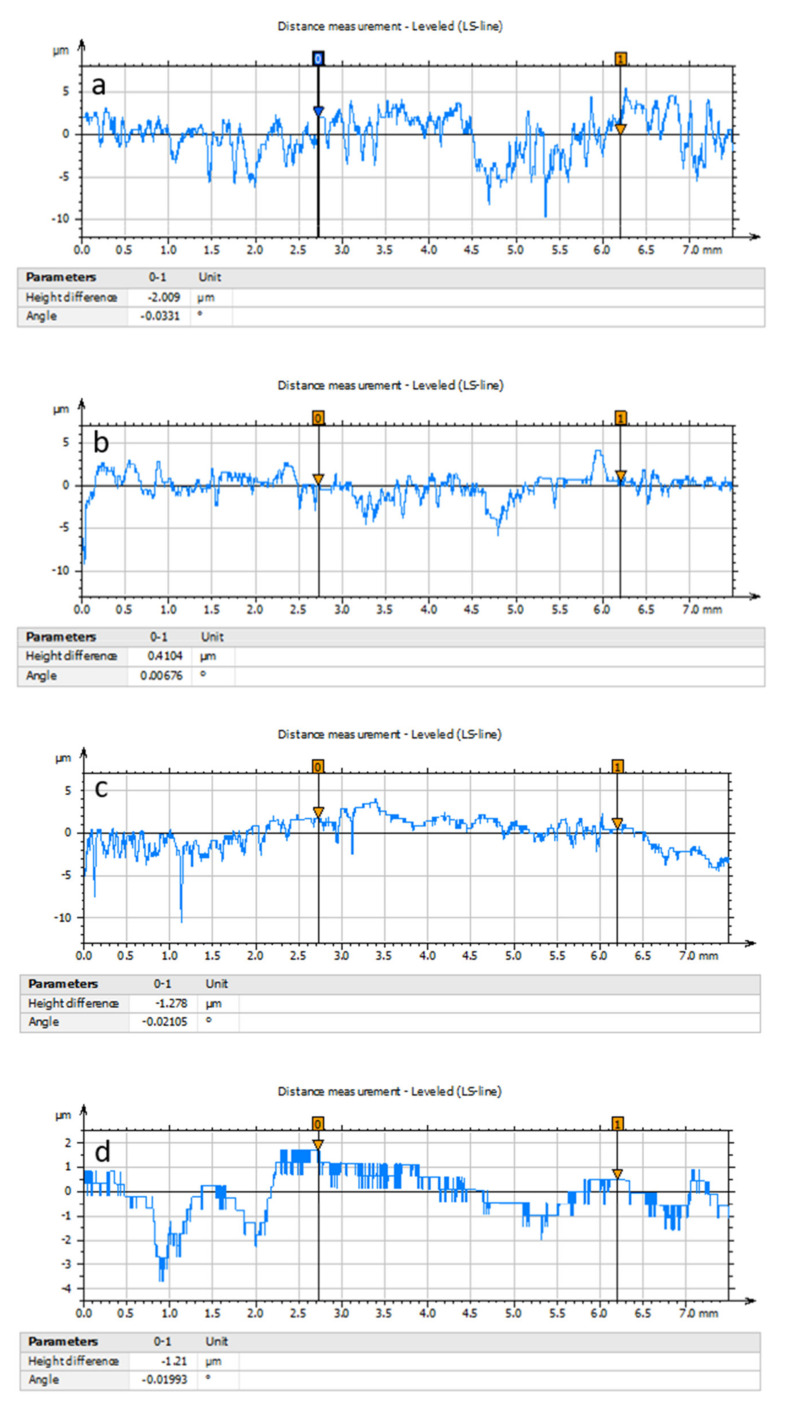
Roughness analysis of the sample at (**a**) simultaneous motion of pin and disc at dry condition, (**b**) reciprocating motion of pin at dry condition, (**c**) simultaneous motion of pin and disc at lubricated condition, (**d**) reciprocating motion of pin at lubricated condition.

**Figure 26 materials-14-05732-f026:**
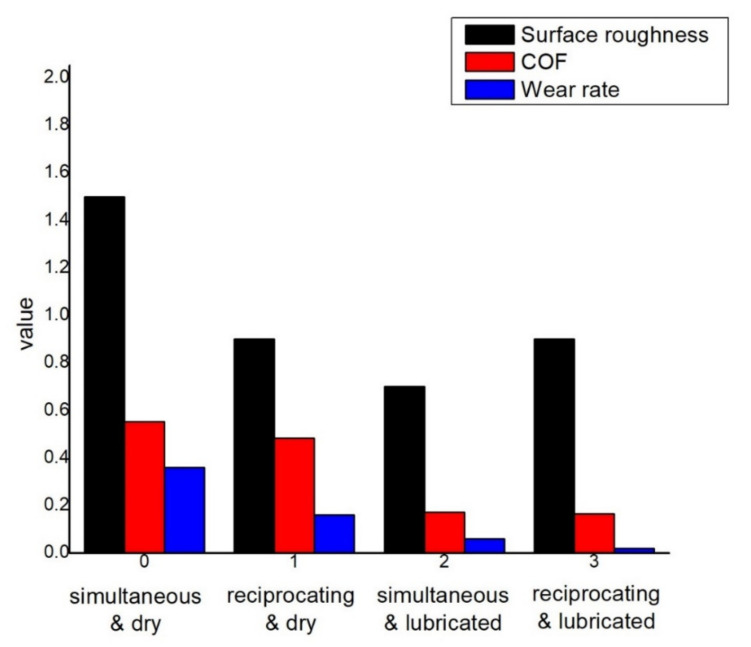
Surface roughness, COF, and wear rate comparison.

**Table 1 materials-14-05732-t001:** Comparison of variation of COF at different conditions.

SL.	Friction at 3.5 N Applied Force, 0.45 m/s Disc Velocity, 0.15 m/s Pin Velocity, and without Lubricant	Comparison of Friction at 3.5 N Applied Force, 0.15 m/s Pin Motion, and without Lubricant with the First One (%)	Comparison of Friction at 3.5 N Applied Force, 0.45 m/s Disc Motion, 0.15 m/s Pin Motion, and with Lubricant with the First One (%)	Comparison of Friction at 3.5 N Applied Force, 0.15 m/s Pin Motion, and with Lubricant with the First One (%)
1	0.18961	−5.273983	−13.13749	−16.9664
2	0.18961	−5.273983	−12.27783	−16.18058
3	0.19125	−5.228758	−13.03007	−16.72157
4	0.19125	−4.371242	−12.17255	−16.12026
5	0.19289	−5.184302	−12.06905	−16.83343
6	0.19452	−5.14086	−11.96792	−16.76434
7	0.19289	−3.489035	−11.22401	−16.06097
8	0.19452	−3.459798	−11.12482	−16.76434
9	0.19616	−4.266925	−11.86786	−16.70065
10	0.19616	−3.430873	−12.70392	−16.70065
11	0.19616	−2.594821	−12.70392	−17.46024
12	0.19779	−3.397543	−14.24743	−17.38713
13	0.19616	−3.430873	−13.53487	−15.93597
14	0.19452	−3.459798	−13.64898	−16.76434
15	0.19452	−3.459798	−14.49208	−16.76434

**Table 2 materials-14-05732-t002:** Wear rate comparison at different conditions.

SL.	Wear Rate at 3.5 N Load, 0.45 m/s disc, 0.15 m/s Pin Velocity, and Dry Condition	Variation of Wear Rate at 3.5 N Load, 0.15 m/s Pin Motion, and Dry Condition with the First One (%)	Variation of Wear Rate at 3.5 N Load, 0.45 m/s Disc Motion, 0.15 m/s Pin Motion, and with Lubricant with the First One (%)	Variation of Wear Rate at 3.5 N Load, 0.15 m/s Pin Motion, and with Lubricant with the First One (%)
1	0.45	−62.88889	−86.66667	−95.55556

**Table 3 materials-14-05732-t003:** COF Comparison at different conditions.

SL.	Friction at 2.5 N Load, 0.45 m/s Disc, 0.25 m/s Pin Velocity, and No Lubricant	Comparison of Friction at 2.5 N load, 0.25 m/s Pin Velocity, and No Lubricant with the First One (%)	Comparison of Friction at 2.5 N Applied Force, 0.35 m/s Disc, 0.15 m/s Pin Velocity, and No Lubricant with the First One (%)	Comparison of Friction at 2.5 N Load, 0.15 m/s Pin Velocity, and No Lubricant with the First One (%)
1	0.25283	−33.38211	−5.695527	−39.07764
2	0.25503	−33.09415	−5.646395	−38.74054
3	0.25723	−33.27761	−5.598103	−38.40921
4	0.25723	−32.8111	−5.598103	−38.40921
5	0.25943	−33.38087	−5.55063	−38.77732
6	0.25943	−32.53286	−5.55063	−38.85441
7	0.26163	−32.4657	−5.503956	−39.29213
8	0.26163	−32.2593	−5.503956	−39.36857
9	0.26383	−31.9903	−7.125801	−39.94997
10	0.26163	−32.2593	−7.18572	−40.2859
11	0.26163	−32.2593	−8.026602	−41.12678
12	0.25943	−32.53286	−8.094669	−41.47554
13	0.25943	−33.38087	−8.942682	−41.47554
14	0.25723	−32.8111	−9.019166	−41.83027
15	0.25503	−33.09415	−8.234325	−41.32847

**Table 4 materials-14-05732-t004:** Variation of wear rate at distinct operating conditions.

SL.	Wear Rate at 2.5 N Applied Force, 0.45 m/s Disc Velocity, 0.25 m/s Pin Velocity, and Dry Condition	Comparison of Wear Rate at 2.5 N Applied Force, 0.25 m/s Pin Velocity, and Dry Condition with the First One (%)	Comparison of Wear Rate at 2.5 N Applied Force, 0.35 m/s Disc Velocity, 0.15 m/s Pin Velocity, and Dry Condition with the First One (%)	Comparison of Wear Rate at 2.5 N Applied Force, 0.15 m/s Pin Velocity, and Dry Condition with the First One (%)
1	0.47	−63.82979	−25.53191	−68.08511

**Table 5 materials-14-05732-t005:** Comparison of COF at different conditions.

SL.	Friction at 1.5 N Load, 0.45 m/s Disc, 0.2 m/s Pin Velocity, and Dry Condition	Variation of Friction at 1.5 N Load, 0.2 m/s Pin Velocity, and Dry Condition with the First One (%)	Variation of Friction at 4.5 N Load, 0.45 m/s Disc, 0.2 m/s Pin Velocity, and Dry Condition with the First One (%)	Variation of Friction at 4.5 N Load, 0.2 m/s Pin Velocity, and Dry Condition with the First One (%)
1	0.52566	−9.939885	−61.45607	−71.53864
2	0.53028	−9.853285	−61.79188	−71.7866
3	0.53491	−9.769868	−61.81601	−71.72422
4	0.53953	−11.39881	−62.14298	−71.66237
5	0.54416	−13.00169	−62.1637	−71.90348
6	0.54416	−11.164	−62.04793	−71.60394
7	0.54878	−12.75557	−62.48223	−71.54415
8	0.55341	−12.51333	−62.50158	−71.48769
9	0.55341	−12.51333	−62.20524	−71.48769
10	0.55341	−13.34996	−62.20524	−71.19134
11	0.54878	−13.46077	−61.88637	−70.64944
12	0.54416	−12.72604	−61.26323	−70.40025
13	0.54416	−13.57689	−61.56278	−70.70163
14	0.53953	−13.69155	−61.53689	−70.75417
15	0.53491	−12.9461	−61.20469	−70.50158

**Table 6 materials-14-05732-t006:** Comparison of wear rate at different conditions.

SL.	Wear Rate at 1.5 N Load, 0.45 m/s Disc Velocity, 0.2 m/s Pin Velocity, and Dry Condition	Comparison of Wear Rate at 1.5 N Load, 0.2 m/s Pin Velocity, and Dry Condition with the First One (%)	Comparison of Wear Rate at 4.5 N Applied Force, 0.45 m/s Disc, 0.2 m/s Pin Velocity, and Dry Condition with the First One (%)	Comparison of Wear Rate at 4.5 N Load, 0.2 m/s Pin Velocity, and Dry Condition with the First One (%)
1	0.36	−55.55556	+73.6111	−33.33333

## Data Availability

Data is contained within the article.

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
