# Peer review of "Effects of Self-Lubricant Coating and Motion on Reduction of Friction and Wear of Mild Steel and Data Analysis from Machine Learning Approach"

_materials, 2021, doi:10.3390/ma14195732_

Round 1
Reviewer 1 Report
Authors conducted this research to evaluate Effects of Self-Lubricant Coating and Motion on Reduction of Friction and Wear of Mild Steel and Data Analysis using Machine Learning Approach. This research focuses on the tribological study of coated mild steel in different conditions.
The paper’s subject could be interesting for readers of journal. Therefore, I recommend this paper for publication in this journal but before that, I have a few comments on the text that should be addressed before publication:
Comments:
1)In the abstract section,line 5: About this phrase "In the experiments" authors used uppercase to write "In". It should be corrected by using lowercase like this "in".
2)In page 4, Figure 3: The position and size of the used picture are not appropriate. Size of the picture should be smaller. About the position of the picture, authors should move it to the right. Also the color of the used arrow on the picture to explain is not good, because it is hard to see that regarding the background's color.
3)About the title of Figures in this article, somewhere authors put the titles at the middle alignment and elsewhere authors put the titles at the left alignment. They have to be integrated in terms of alignment.
4) In Figure 10 and 11, the numbers of vertical and horizontal axes of the chart are too close to the axes. Authors should correct it with more space between them.
5)In this article the authors used the word "We" too much and it is not right and appropriate. For example authors could use third person subjects and verbs to avoid using "We" repeatedly.
6)In the conclusion section, the authors did not mention anything about conflict of interests or research funding. Also there is no suggestion about future studies with similar titles.
7)Which softwares have been used to draw and export the charts?. Also authors should explain why they selected used softwares in this work over other softwares.
8) Since recently it has been proved that artificial intelligence (AI) and machine learning has a numerous applications in all of engineering fields, I highly recommend the authors to add some references in this manuscript in this regard. It would be useful for the readers of journal to get familiar with the application of AI in other engineering fields. I recommend the others to add all the following references, which are the newest references in this field of computer engineering [1], biomedical engineering [2], petroleum engineering [3]
[1] Tavakoli, S., Hajibagheri, A. and Sukthankar, G., 2017. Learning social graph topologies using generative adversarial neural networks. In International Conference on Social Computing, Behavioral-Cultural Modeling & Prediction.
[2] Tavakoli, S., & Yooseph, S. (2019, November). Algorithms for inferring multiple microbial networks. In 2019 IEEE International Conference on Bioinformatics and Biomedicine (BIBM) (pp. 223-227). IEEE.
[3] Roshani, Proposing a gamma radiation based intelligent system for simultaneous analyzing and detecting type and amount of petroleum by-products,Nuclear Engineering and Technology 53 (4), 1277-1283.
Reviewer 2 Report
Review Comments:
- The abstract should be revised. The significances in engineering field should be highlighted.
- The authors are suggested to explain the novelty of the paper.
- The authors are suggested to add more references related to the paper.
- For Figure 1 and Figure 3, the authors are suggested to redraw it. It is not very clear.
- For the experimental setup, the schematic diagram of the experimental apparatus should be added in the paper.
- For the surface lubrication regime in Section 3.2, further step lubrication regimes analysis should be added. The authors are suggested to add comments as well as the references below. These references clearly illustrate the lubrication regimes as well as the surface roughness effects on the interface.
[1] Theoretical and experimental research on the micro interface lubrication regime of water lubricated bearing. Mechanical Systems and Signal Processing, 2021, 151: 107422.
[2] Numerical analysis of added mass and damping of elastic hydrofoils[J]. Journal of Hydrodynamics, 2020, 32(5): 1009-1023.
[3] An investigation on the lubrication characteristics of floating ring bearing with consideration of multi-coupling factors. Mechanical Systems and Signal Processing, 2022, 162: 108086.
- For the conclusions, it is too long. Several brief points of conclusions are enough. The authors are suggested to rewrite the conclusions.
- The English should be improved considerately.
Major revision.
Round 2
Reviewer 1 Report
All the comments have been addressed correctly and dthe paper is ready for publication in the present form.
Reviewer 2 Report
The authors have revised the paper according to the review comments carefully point by point. It can meet the requirements of the journal now. Accept it in the present form.